# Generalizing Microscopy Image Labeling via Layer-Matching Adversarial Domain Adaptation

**Vishrut Goyal** [1]  **Michael Liu** [2]  **Andrew Bai** [2]  **Neil Lin** [3]  **Cho-Jui Hsieh** [2]

## Abstract

Image-to-image translation models are valuable tools that convert light microscopy images into in silico labeled biological immunofluorescence images, enabling non-invasive and label-free measurement of protein expression levels in live cells. Despite their potential, these models have not gained significant traction in the life sciences due to their low transferability and lack of robustness to common variances in microscopy images. Additionally, re-training a model for each new microscope setting is infeasible due to the high cost of data acquisition. In this work, we explore domain adaptation techniques to make image-to-image translation models more robust to common distribution shifts in microscopy images. Specifically, we propose Layer-Matching Adversarial Domain Adaptation (LM-ADDA), a general framework that leverages the information-rich latent spaces within the translation model to perform unsupervised domain adaptation. Through experiments on multiple domain shifts, we demonstrate that LM-ADDA enhances the robustness of image-to-image translation models without requiring additional paired or labeled data.

## 1. Introduction

Transmitted light microscopy is a fundamental tool in the life sciences, allowing researchers to analyze cellular identity and functions through the examination of cell morphology. Machine learning techniques, such as image-to-image translation, have effectively transformed transmitted light microscopy images to extract previously inaccessible information, such as protein expression levels in live cells. These advancements hold potential applications in assessing the molecular-based characteristics of live cells for therapeutics applications (Imboden et al., 2021; Weber et al., 2023; Imboden et al., 2023).

However, the variability inherent in microscopic settings, such as different microscopy types and settings, pose a significant challenge for the generalization of image-to-image translation models to new experimental settings. This inconsistency leads to a major decline in the performance of in images captured even under slightly altered settings, which complicates the task of analyzing and interpreting data across different microscopes and laboratories. Given the frequent occurrence of these discrepancies, there is a need to adapt image translation models to work reliably across diverse imaging conditions without retraining models from scratch—a process which is infeasible to repeat given the need for a large, paired training dataset in the new setting.

We propose a framework which extends adversarial discriminative domain adaptation to image-to-image translation models and generalizes matching to all layers. The primary contributions of this investigation are summarized follows:

- We apply adversarial discriminative domain adaptation to image translation models.
- We propose a framework called Layer-Matching ADDA (LM-ADDA), which adversarially aligns the new distribution to the source distribution at a given model layer.
- This is the first work studying the domain adaptation problem on real-world microscopy images of mesenchymal stromal cells under real-wolrd domain shifts. We show the effectiveness of LM-ADDA and evaluate the impact of the layer at which matching occurs.

A key takeaway from our initial findings is the counterintuitive observation that "less is more" in two critical aspects of domain adaptation: matching early layers with a smaller model configuration often yields better results than attempting to do the traditional final layer match, and focusing the adaptation on a single layer proves more effective than distributing matching across multiple layers. These insights provide a new perspective about the depth of adaptation required and opens new avenues for efficient model tuning

[1]Jericho High School [2]Department of Computer Science, University of California Los Angeles [3]Department of Bioengineering, University of California Los Angeles. Correspondence to: Vishrut Goyal <vishrut.goyal@jerichoapps.org>.

*Accepted at the 1st Machine Learning for Life and Material Sciences Workshop at ICML 2024.* Copyright 2024 by the author(s).

in the lieu of domain shifts.

## 2. Related Work

**Image translation for Virtual Labeling**   Image-to-image translation models have been widely used in biomedical data for various purposes, including image modality conversion (Ghahremani et al., 2022; Chen et al., 2021), image restoration (Fang et al., 2021), and image segmentation (Ma et al., 2024). One common application is translating structural MRI scans to positron emission tomography (PET) since MRI is much cheaper, less invasive, and more available than PET (Vega et al., 2024; Chen et al., 2024). Another common application is translating microscopy cell images to fluorescence images to observe specific features of the cells without destructive staining (Hu et al., 2022; Lee et al., 2021; Imboden et al., 2021). This allows analyzing multiple fluorescence copies of the same microscopy image simultaneously, which is previously impossible since marker staining is mutual exclusive. The image-to-image translation models can be trained in a supervised, in which paired image data is hard to acquire, or unsupervised manner, where significantly more data is required even with recent advances in generative modeling under limited data regime (Bourou et al., 2023). The high training and data collection cost highlight the importance of robustness of the translation models, since training a new model for every slightly-perturbed input data distribution is infeasible.

**Previous work on domain adaptation**   Domain adaptation for deep neural nets has been a well-explored problem. There are two major approaches for this problem. The first is at the image stage, by explicitly converting images in the new domain to the original domain, where the original model can be applied, such as current works for image translation models (Chen et al., 2020). Our work follows the second approach, at the feature stage, in which the distributions from both domains are matched in some latent space (Huang et al., 2006; Long et al., 2015; Tzeng et al., 2017). Goodness of match can be captured with pre-defined metrics that measure distance between domain distributions, including maximum mean discrepancy (Wang et al., 2020) and clustering-based objectives (Barbato et al., 2021). The match can also be measured with an adversarial loss, where a discriminator is trained jointly with the domain adaptor to provide signal for differences between domains. This approach works without constraints on the feature space used. The minimax objective of the discriminator and domain adapter can be implemented with GAN-style iterative optimization (Tzeng et al., 2017) or loss gradient reversal (Ganin & Lempitsky, 2014; Ganin et al., 2015).

**Domain adaptation for image-to-image translation**   Existing literature for domain adaptation in image-to-image translation tasks focus on improving the model architecture (Hoffman et al., 2018) or utilizing additional labeled information to align the distributions (Mütze et al., 2022). Yan et al. (2019) proposed an multi-source domain adaptation framework that first translates a given input into one specific source domain, and then reuses a pretrained supervised model to map the input to the target domain. However, these methods all implicitly assumed that domain adaptation should be performed in the pixel-space for image-to-image translation tasks. In this work, we investigate the selection of joint latent space to perform domain adaptation and find that contrary to existing belief, adaptation in an intermediate model layer latent space can outperform the output space.

## 3. Methodology

### 3.1. Problem definition

The goal of image translation is to label an image $x \in X$ with a corresponding output image $y \in Y$. In order to do this, we train a model $G : X \to Y$. In our case, $X$ is the set of phase-contrast microscopy images, while $Y$ represents immunofluorescence images which show marker protein levels. Given that immunofluorescence intensity represents the density of the protein, this can be seen as labelling the pixels of the input image with protein level. Our $G$ is the UNet architecture (Ronneberger et al., 2015), which includes convolutional downsampling to a bottleneck, followed by upsampling to the original shape. In order to preserve lower level information, skip connections in the front half directly link to the back half as well.

However, in the real world, we may wish to work with a new input distribution $X'$, which has the same dimension as $X$ : for example, bright-field rather than phase-contrast microscopy. Without access to any paired images in the new input domain, it is infeasible to train a new model from scratch. The goal of domain adaptation is to train a model $G' : X' \to Y$ based on a previously trained $G$.

For the image translation domain adaptation problem, we propose the following Layer-Matching Adversarial Discriminative Domain Adaptation (LM-ADDA) algorithm. Similar to the Adversarial Discrimative Domain Adaptation (ADDA) algorithm (Tzeng et al., 2017), our goal is to match the final output distribution of the original model with the new model. The natural way to do this is to adversarially match the final layer of the image-to-image translation model. Our key contribution is to consider matching at internal layers.

One way to intuitively think about how matching at an earlier layer could be superior to matching the final objective is in terms of the information latent space. In an ideal scenario, an image-to-image translation model can be seen as a two-step process: Extracting features from the image, and

converting these features into an output image. The domain shift in the microscopy problem setting should preserve the underlying semantic information, so that the second step of converting features to the translated image can remain unchanged. This view is particularly important in our setting since the inputs x and x' do not have any paired correspondence. Hence, all that is needed is to identify an optimal latent space representing the right features for the task to match. Matching the later layers from the model than the optimal one can lead to spurious information induced by image-to-image translation framework beyond the features we need to align, thus reducing the signal to noise ratio and making matching difficult. Thus, matching the features which are directly affected by the domain shift can require matching at representation at layers other than the final one. First, we translate both $X$ and $X'$ to a common representation, and then, we decode this representation to the final output space. Rather, we define a layer $l$ of $G$ as the shared space, and learn a new function $G'_l : X' \rightarrow G_l(X)$. This function is composed with the unchanged layers after $l$ in $G$ to create $G'$, which goes from $X'$, to the shared space, to $Y$.

We formulate the problem of minimizing discrepancy between the distributions of the two domains at a given layer as a generative adversarial network (GAN) (Goodfellow et al., 2020). To deem the two latent representations as matched, they must be indistinguishable. Thus, we define a network $D$ as the domain discriminator, tasked to classify a sample as either $G_l(x)$ or $G'_l(x')$. Concurrently, the generator $G'_l$ plays a minimax game against $D$, selecting parameters $theta$ within the first $l$ layers to minimize the maximal possible difference in the predictions between $G_l(x)$ and $G'_l(x')$.

Mathematically, the adversarial loss can be written as

$$\mathcal{L}_{\text{adv}} = \mathop{\mathbb{E}}_{x \sim X} \left[ \log(D(G_l(x))) \right] - \mathop{\mathbb{E}}_{x' \sim X'} \left[ \log(D(G'_l(x', \theta))) \right]$$

and the discriminator and generator are utilize the following minmax objective

$$\min_{G'} \max_{D} \mathcal{L}_{\text{adv}}(D, G').$$

Beyond applying ADDA to the image translation application, our main contribution is demonstrating that matching the final layer may not yield the best results when conducting domain adaptation in image translation. We provide the reasoning below and present empirical evidence showing that matching intermediate layers is more effective than the final layer in our experiments.

# 4. Results

## 4.1. Experimental Settings

We run our experiments with the task of labelling light-microscopy images with corresponding fluorescence mi-croscopy images stained for a given marker. For this, we utilized human mesenchymal stromal cells (MSCs) as our model. MSCs are stem cells that can differentiate into a variety of cell types such as osteoblasts and myocytes. These cells are commonly studied in the field of regenerative medicine because of its versatile differentiation capabilities and its potential in organ transplantation. Thus, the ability to predict and identify what different proteins are present in MSCs is critical for advancing these cell-based therapeutic strategies.

Our dataset of images were taken with a spatial resolution of 100 μm, which provided enough detail analyze the cellular structures of the MSCs. Our source models are trained on 273 of these paired images, and we reserved 50 images for testing.

We used the 16 layer 1024x1024 UNet and 70x70 patch CNN from Pix2Pix (Isola et al., 2017) as the generator $G$ and discriminator $D$ respectively. Learning rates from $10^{-5}$ to $10^{-1}$ were scanned. Those which saw no mode collapse, i.e., generator loss remain bounded, across all layers were selected.

We compare the final performance of the model on the shifted input dataset, depending on which layer the matching occurred. The metric used is Pearson's Correlation, computed between pixel intensities for the input and output pixel at the same location. This is suited to this task because the accuracy with which marker levels are predicted at a given location is more important than the generalized appearance of the image. Our baseline is the performance of the unadapted model applied directly to the domain shifted input.

## 4.2. Domain shift from Phase Contrast to Bright Field

We first consider the domain shift from Phase Contrast to Bright Field microscopy for the input images. Phase Contrast microscopy is often used to enhance the contrast of transparent and colorless specimens. It is particularly useful for viewing details in cells that would otherwise be difficult to see. On the other hand, Bright Field microscopy illuminates the sample directly with brightlight and is usually used with stained markers on the cells. This method provides less contrast and detail in unstained cells, making it challenging to discern subtle cellular features. This contrast in very common imaging techniques shows the challenges in adapting models trained on Phase Contrast images to perform well on Bright Field images, where the visual information is fundamentally different. Some examples of images in Phase Contrast (PC) domain and Brightfield (BF) domain are visualized in Figure 1 and Figure 2.

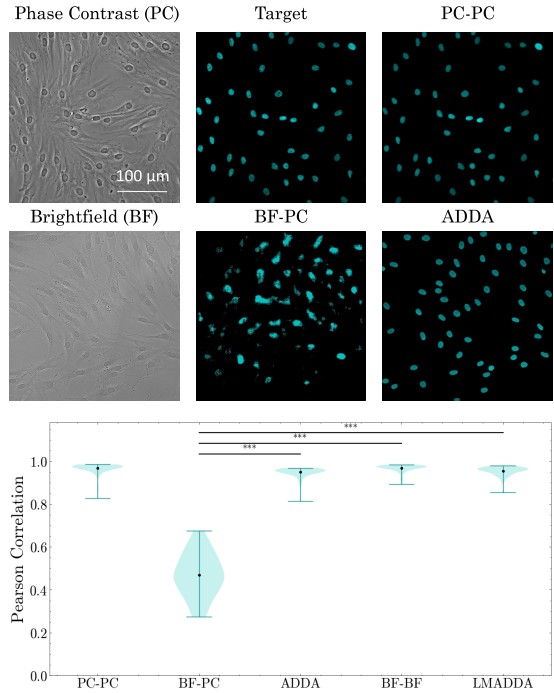

*Figure 1. Evaluation of domain adaptation performance on DAPI stain.* Top panel: Comparison across microscopy techniques. Bottom panel: Pearson correlation analysis.

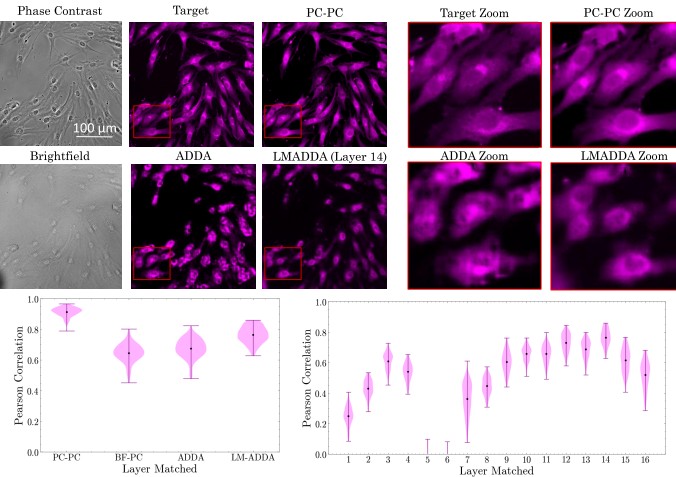

*Figure 2. Evaluation of domain adaptation performance on complex biological marker CD29.* Top panel: Comparison across microscopy techniques including zoomed regions for detailed comparison. Bottom panels: Pearson correlation coefficients across different transformation methods (left) and correlation by layer matched (right).

### 4.2.1. CELL NUCLEUS IDENTIFICATION

The first case we examine is the DAPI stain which identifies the cell nucleus, a relatively simple task. We apply LM-ADDA to this setting and use bright-field images in the target domain. Figure 1 presents the results and some sampled images. We observe that both LM-ADDA and ADDA can achieve near perfect results.

For domain adaptation methods, we observe LM-ADDA slightly outperforms ADDA. We observe that for this simple task, matching on almost any layer is able to achieve good performance. This task proves that matching on a later layer improves results compared to matching the final output. Furthermore, directly running the source model on the target domain (BF-PC) leads to a huge performance loss, demonstrating the importance of domain adaptation.

### 4.2.2. CD29 MARKER

On the other hand, the need for LM-ADDA in this setting is clearly shown with the CD29 marker. CD29 is a crucial protein that plays a role in cell signaling and differentiation in stem cells. They are commonly found around the nucleus. We also consider the domain shift from PC to BF and plot the sampled images and our experimental results in Figure 2.

Here, the shift remains the same, however the output dis-

tribution is far more complex than simply identifying the nucleus. We observe LM-ADDA significantly outperforms ADDA with a Pearson of 0.76 instead of 0.65. Qualitatively, matching at different UNet layer best recovers different features. Matching on layer 13, we see that the halo around the nucleus, a key property of CD29, is strongly preserved.

### 4.3. Adapting to Overexposed Images

Next, we examine another domain shift: overexposure, an important issue that occurs in real-world microscopy settings. In particular, we investigate whether greater magnitude shifts affect the optimal layer for matching. For this, we create samples with different overexposure levels and compare our methods with baselines. The results and sampled images are shown in Figure 3.

We see that the model is robust to small overexposures without any adaptation, but declines thereafter. Applying LM-ADDA to layer 2 significantly improves the performance, especially for larger overexposure magnitudes. We note successful results matching layers in the beginning, middle, and end of the model, with a noticeable decline in the middle of the first and second halves. Specifically, we see the best performance in layer 2, a trend which remains for all levels of overexposure. Given that this is a simple, pixel level shift, undoing it in the first couple of layers seems intuitively reasonable.

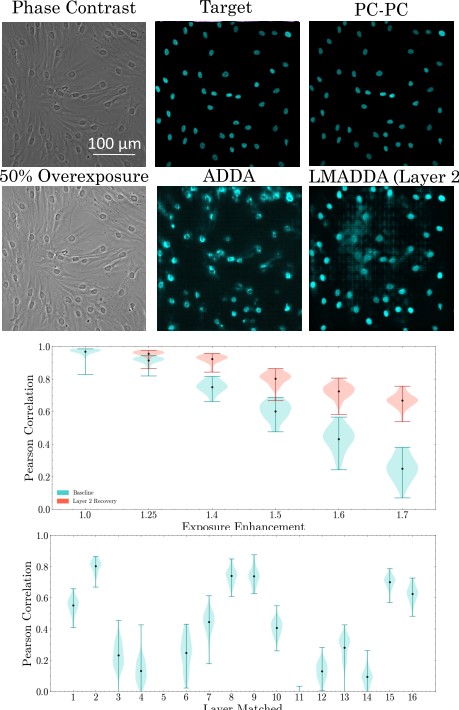

Figure 3. *Analysis of domain adaptation performance under varying levels of overexposure.* Top panel: Comparison across overexposure. Bottom panels: Pearson correlation coefficients across exposure enhancement levels (top graph) and by layer matched (bottom graph) when overexposed 150%.

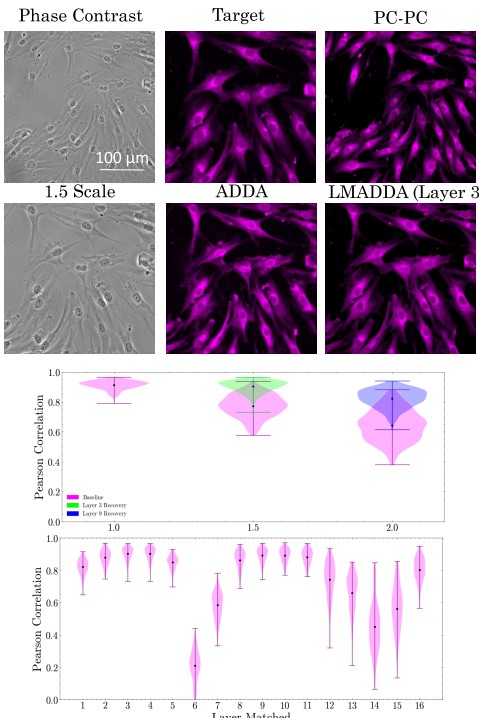

Figure 4. *Analysis of domain adaptation performance under varying levels of scaling.* Top panel: Comparison of overexposure. Bottom panels: Pearson correlation coefficients across scale enhancement levels (top graph) and by layer matched when scaled 150% (bottom graph).

## 4.4. Adapting to Scaled Images

Finally, we examine the task of scaling. This is unique, because the output distribution is scaled as well, changing the problem setting of having the same output distribution. To imagine successful adaptation here would be to imagine that there is a scale invariant latent space, which can put both zoomed in and out images into the same language, and then turn them back into their original scaling. The results are shown in Figure 4. We see that the model is robust to shifts even up to 1.5x, on both CD29 and Nucleus. Past this, domain adaptation is reasonably successful. We see the best results in the central layers, supporting the hypothesis that such a latent space may exist in the deepest layers.

## 5. Conclusion

We find that adversarial domain adaptation can successfully adapt microscopy image translation methods across a variety of common real-world shifts, and across varying shift magnitudes. We show that the layer at which matching occurs has a significant impact on ability to undo shifts, and that the optimal layer varies depending on the nature of the domain shift.

We note some limitations of our work as well. Currently, the unpaired matching still requires a large number of input images from the target domain, which can be expensive in a real-world setting. In addition, the framework assumes a minor shift, given that we start with existing weights, and attempt to use the same feature space. Finally, discovering the optimal layer requires testing all layers, although this can be done relatively inexpensively.

Further work can work explore combining information from multiple layers, which has shown some promising results (Barbato et al., 2021). We can also attempt to explain the way performance varies through successive model layers and why matching certain features best undoes the domain shift, in order to theoretically find an optimal layer given a certain model and shift.

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
