# OpenReview forum: "Generalizing Microscopy Image Labeling via Layer-Matching Adversarial Domain Adaptation"
_ICML.cc/2024/Workshop/ML4LMS — ML4LMS Poster_

### Official Review · Reviewer_PNX5 · 2024-06-11
**Domain Adaptation in Image to Image Translation Models**

**Rating:** 6
**Confidence:** 4

**Review:**

In the paper the authors are interested in translating phase contrast microscopy images to immunofluorescence images with the goal of adapting the above trained CNN model to be also useful on translating bright field microscopy images to immunofluorescence images. The quality of the work is fair. The main contribution of the authors is they show training on intermediate layers rather than on the final output layer leads to better translational capabilities. The above method performs on par or slightly better than the ADDA method. However analysis to the ADDA method is missing in the overexposed and scaled image cases. The clarity in the text could be improved at several locations and are listed below :

  - On page 2 line 83 the authors have not mentioned what is the second approach.
  - On page 2 line 98 there is some in consistency in the text where the authors mention that both $X$ and $X'$ are converted to a shared representation but the formula, $G_l' : X' \to G_l(X)$ indicates a conversion of $X'$ to $X$.
  - The authors mention several ways of minimizing the difference in the data distributions but do not provide more context why they chose to use the discriminator - generator approach.
  - Inclusion of more details on training and diagrams would help the reader to understand the method better.
  - The authors do not mention the utility of bright field microscopy over phase contrast and why it would make sense to do bright field first rather than just use phase contrast microscopy for all the samples.
  - The authors do not comment on why at certain layers there is a drop in the PCC values.

---

### Official Review · Reviewer_8eDM · 2024-06-12
**Interesting paper**

**Rating:** 7
**Confidence:** 4

**Review:**

Overall, the work seems interesting. A couple questions:
  - What happens if the new domain should not actually have the same distribution? For example, consider malignant cells with unusual nuclei, etc.? In this case, the cells are actually different. It is not the same cells but with bright field in stead of phase contrast.
  - In Fig 1, is the tissue sample the same in all cases?

---

### Official Review · Reviewer_vSsR · 2024-06-12
**Accept - Good paper that extends adversarial discriminative domain adaptation in a layer-wise manner to image-to-image translation models**

**Rating:** 8
**Confidence:** 5

**Review:**

A well written paper that extends the work on adversarial discriminative domain adaptation to image-to-image translation. The authors also explore applying adversarial domain transfer at different intermediate layers of the model. Using their layer-matched ADDA methodology, they observe that matching early layers produces better results than domain matching at the final layer for translation tasks and provide an intuitive reasoning for their observation.

The motivation, problem definition and methodology sections are well written. The experiments and results were conducted on limited datasets, but are sufficient to demonstrate the methodology for initial exploration.